# ADVANCING ENERGY EFFICIENCY IN ON-DEVICE STREAMING SPEECH RECOGNITION

## ABSTRACT

Power consumption plays a crucial role in on-device streaming speech recognition, significantly influencing the user experience. This study explores how the configuration of weight parameters in speech recognition models affects their overall energy efficiency. We found that the influence of these parameters on power consumption varies depending on factors such as invocation frequency and memory allocation. Leveraging these insights, we propose design principles that enhance on-device speech recognition models by reducing power consumption with minimal impact on accuracy. Our approach, which adjusts model components based on their specific energy sensitivities, achieves up to 47% lower energy usage while preserving comparable model accuracy and improving real-time performance compared to leading methods.

## 1 INTRODUCTION

Automatic streaming speech recognition (streaming ASR) is a crucial technology that enables real-time transcription of user speech into text, typically requiring latency under 500 milliseconds. This technology is essential for various applications on mobile and wearable devices. The seamless integration of streaming ASR into devices such as smartphones and VR/AR glasses enhances user interaction, supporting functionalities such as interface navigation, voice commands, real-time communication, and accessibility features. Speech recognition and on-device AI remain important topics in machine learning conferences including ICLR, for example, (Yao et al., 2024; Chen et al., 2024; Hu et al., 2024; Haliassos et al., 2023; Chang et al., 2023; Shi et al., 2022; Qiu et al., 2022; Kim et al., 2022; Variani et al., 2022; Leng et al., 2021; Stephenson et al., 2019; Chorowski et al., 2015; Zhao et al., 2023; Lin et al., 2021; Cai et al., 2020).

Despite its critical role, the deployment of on-device streaming ASR faces a significant challenge in power consumption. High energy demand can severely limit the practical usability of devices, necessitating frequent recharges that degrade the overall user experience. Therefore, enhancing the energy efficiency of on-device streaming ASR is crucial.

We explore the intricacies of on-device streaming ASR models with a focus on the Neural Transducer (Graves, 2012). The Neural Transducer represents a class of speech recognition models comprised of three key components: an Encoder for acoustic modeling, a Predictor for language modeling, and a Joiner that combines the outputs of the Encoder and Predictor (see Figure 1). It has emerged as the de facto standard solution for on-device streaming ASR (Graves et al., 2013; He et al., 2019; Li et al., 2021), primarily due to its exceptional balance between computational efficiency and model accuracy. Transformer-based designs have been widely adopted within Neural Transducer models (Shi et al., 2021; Moritz et al., 2020; Dong et al., 2018; Zhang et al., 2020; Yeh et al., 2019; Gulati et al., 2020; Wang et al., 2020; Karita et al., 2019). In our comprehensive analysis, we train and evaluate over 180 Neural Transducer models[1], experimenting with different architectures (e.g., Emformer (Shi et al., 2021), Conformer (Gulati et al., 2020)) and varying the sizes of their core components. This large-scale evaluation sheds light on how these components influence the model's accuracy, real-time factor (RTF),[2] and power consumption.

---

[1]Each model requires 20-40 hours of training on 32 V100 GPUs.

[2]RTF represents the ratio of model inference time to the actual duration of the speech segment being processed. A lower RTF indicates greater efficiency in model inference, demonstrating faster processing.

Our analysis reveals several new findings. First, we discover that the model's energy usage is primarily influenced by the memory traffic associated with loading model weights, which is in turn affected by the invocation frequency of model components and their placement within the device's memory hierarchy. Secondly, there is a remarkable disparity in the invocation frequency of model components, with the Joiner being summoned significantly more often than the Predictor, which in turn is more frequently invoked than the Encoder. Consequently, although the Joiner constitutes only 5-9% of the ASR model's size, it accounts for 48-73% of the model's power consumption. Thirdly, we uncover an intriguing exponential relationship between the ASR model's accuracy and its encoder size. This finding suggests new avenues for research and development in the field of on-device streaming ASR.

Leveraging these insights, we propose a differentiated compression strategy for ASR model components to optimize energy efficiency while minimizing impact on model accuracy. This strategy accesses the power and accuracy sensitivity of each component, considering their invocation frequency and memory placement. We prioritize compression for components that show higher power sensitivity but lower accuracy sensitivity. Compressing these components significantly reduces energy usage while only slightly affecting accuracy. Therefore, our focus is on compressing the Joiner first, followed by the Predictor and the Encoder, and aiming to store the Joiner's weight parameters in energy-efficient local memory. Experiments on LibriSpeech (Panayotov et al., 2015) and Public Video datasets demonstrate that our strategy achieves a significant reduction in energy usage by up to 47% and a notable decrease in RTF by up to 29%, all while preserving similar model accuracy when compared to the state-of-the-art compression strategies. These prior strategies often overlook the diverse runtime characteristics of ASR model components, highlighting our method's efficiency in using these distinctions.

This paper presents the following contributions:

- Through a comprehensive analysis of the power consumption associated with on-device streaming speech recognition, we have identified several key findings. Notably, we discovered that the energy consumption of individual ASR model components is influenced not only by their respective model sizes but also by the frequency with which they are invoked and their memory placement strategies. This insight challenges the conventional wisdom that larger model components inherently consume more energy, highlighting the importance of considering operational dynamics and memory management in energy consumption.

- We have developed a set of design guidelines aimed at enhancing the energy efficiency of on-device streaming speech recognition systems. The implementation of these guidelines has demonstrated a substantial reduction in energy consumption—by up to 47%—and RTF—by up to 29%—while maintaining similar model accuracy when compared to the state-of-the-art methods.

- Our study reveals an exponential relationship between on-device streaming ASR's model accuracy and encoder size, indicating diminishing returns on accuracy with larger encoders. Our findings encourage the community to reconsider current approaches and more efficiently use computational resources and memory in on-device streaming ASR.

## 2 BACKGROUND

### 2.1 NEURAL TRANSDUCER: ON-DEVICE STREAMING ASR

The Neural Transducer architecture, also known as RNN-T, first introduced in (Graves, 2012), is the state-of-the-art solution to on-device, streaming speech recognition (Graves et al., 2013; He et al., 2019; Li et al., 2021). The Neural Transducer models the alignment between audio and text (Prabhavalkar et al., 2024), integrating a compact language model and an acoustic model within a single framework. Its design effectively reduces its memory footprint, rendering it exceptionally suitable for devices with limited resources (Shangguan et al., 2019; Venkatesh et al., 2021). The Neural Transducer has a short latency, meeting the requirement for streaming speech recognition that latency should typically be less than 500 milliseconds. To the best of our knowledge, most leading companies in the industry are using Neural Transducer models as their go-to choice for on-device streaming speech recognition (Li et al., 2024; Le et al., 2023; Wang et al., 2023; Radfar et al., 2022).

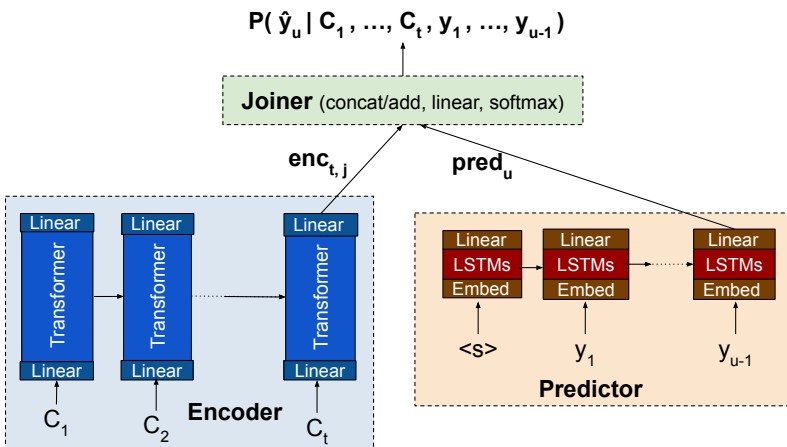

$$P(\hat{y}_u \mid C_1, ..., C_t, y_1, ..., y_{u-1})$$

Figure 1: A schematic representation Transformer-based Neural Transducer. While most of the weights are in the Encoder, the Joiner is called most frequently during inference.

The Neural Transducer architecture comprises three main components: an Encoder, a Predictor, and a Joiner, as depicted in Figure 1. The Encoder processes acoustic feature inputs by receiving chunks of audio $(C_1, ..., C_t)$, where each chunk $(C_t)$ represents a fixed duration of consecutive audio frames $(\mathbf{x}_{t,1}, ..., \mathbf{x}_{t,n})$. These frames are characterized by 80-dimensional log Mel-filterbank features, which are derived using a sliding audio window of 25 milliseconds and a step size of 10 milliseconds. Each frame $(\mathbf{x}_{t,j})$ is then mapped by the Encoder into an embedding $(\mathbf{enc}_{t,j})$. The Predictor, utilizing previously predicted tokens $(y_1, ..., y_{u-1})$, forecasts the embedding of the next token $(\mathbf{pred}_u)$. The Joiner merges the output embeddings from both the Encoder and Predictor, further processing this combined output through a feedforward neural network and subsequently applying a softmax function to determine the probability distribution of the next token. This process enables the Joiner to model the token's probability distribution over the entire set of sentence-piece targets as well as a "blank" token that signifies the end of a frame's transcription.

Presently, the Encoder in Neural Transducer models is predominantly built as a variant of the Transformer, a trend supported by numerous recent studies (Shi et al., 2021; Moritz et al., 2020; Dong et al., 2018; Zhang et al., 2020; Yeh et al., 2019; Gulati et al., 2020; Wang et al., 2020; Karita et al., 2019). In this work, we respectively implement the Encoder of our Neural Transducer using the Emformer (Shi et al., 2021) and Conformer (Gulati et al., 2020), two streaming Transformer variants, to align with this prevailing trend.[3] Transformer-based approaches allow the Encoder to process frames in a chunk collectively, significantly reducing the frequency of its invocation compared to the Predictor and Joiner, which process frames individually. The Predictor is engaged once for each output token bearing explicit meanings (i.e., a sentence-piece target), whereas the Joiner is called upon not only for tokens with clear meanings but also for the "blank" token. Given that most output tokens are "blank," the Joiner operates far more frequently than the Predictor, highlighting a hierarchy in component invocation frequency where the Joiner is the most frequently used, followed by the Predictor, and then the Encoder.

As Figure 2 shows, mobile and wearable devices come equipped with a variety of processors, including mobile CPUs, GPUs, and specialized hardware accelerators, all designed with energy efficiency in mind. For instance, a neural network hardware accelerator previously highlighted by (Lee et al., 2018) boasts a compute energy efficiency of 5 GOPS/mW (INT8), indicating it consumes merely 1mW to perform 5 billion INT8 operations every second. These processors interact with two primary types of memory: local and off-chip. Local memory, which is often comprised of static random-access memory (SRAM), embedded dynamic random-access memory (eDRAM), or other on-chip DRAM technologies, resides within the processor. This setup allows for swift data access, with read/write operations for 64-byte data taking between 0.5 to 20 nanoseconds and being remarkably energy-efficient; for example, accessing SRAM uses about 1.1 to 1.5pJ per byte (Li et al., 2019). Conversely, off-chip memory, typically based on dynamic random-access memory (DRAM)

---

[3]The choice of Transformer variants does not affect the analysis, findings, or conclusions in this paper.

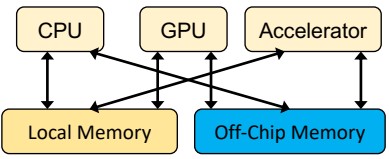

Figure 2: Architecture of mobile and wearable devices.

|  | Encoder | Predictor | Joiner |
|---|---|---|---|
| Size (M) | 60.70 | 8.50 | 4.00 |
| Compute Power (mW) | 0.80 | 0.03 | 0.19 |
| Memory Power (mW) | 47.78 | 12.33 | 57.13 |
| Invocation Frequency (Hz) | 6.25 | 11.53 | 113.50 |

Table 1: A typical model trained on LibriSpeech.

technologies, is slower and less energy-efficient, with 64-byte data read/write operations taking 50 to 70 nanoseconds and consuming around 120pJ per byte (Li et al., 2019).

The contrast in energy efficiency between the processors and both types of memory (local and off-chip) is stark. It leads to a scenario where, for numerous applications such as on-device streaming ASR, memory operations rather than computations become the primary energy drain.

## 2.2 MOBILE AND WEARABLE DEVICES

In this study, we conducted experiments using on-device streaming ASR models on a Google Pixel-5 mobile phone, focusing on measuring RTF and key workload statistics such as the number of operations for the model and the number of invocations for each model component. It is important to note that these workload statistics remain consistent across different device platforms. Given the challenges and potential inaccuracies associated with direct power consumption measurements[4], we opted to model the power consumption for the ASR models operating on mobile or wearable devices, following modeling methodologies established by prior work in speech recognition or computer architecture communities (Li et al., 2024; Micron, 2006; Li et al., 2017; Lee et al., 2009). These devices are assumed to be equipped with a hardware accelerator, 2 MB of local memory, and 8 GB of off-chip memory. The local memory is treated as scratchpad memory, providing flexibility for users to allocate the memory at the model component level. This setup allows for 1.5MB of the local memory to be dedicated to model weights and 0.5MB for intermediate activations. We model the ASR's computing power and memory power based on the platform-independent workload statistics. For computing power, we rely on a compute energy efficiency metric of 5 GOPS/mW for INT-8 operations, as identified by (Lee et al., 2018), and for memory power, we apply energy efficiency figures of 1.5pJ/byte for local memory and 120pJ/byte for off-chip memory, according to (Li et al., 2019). This approach enables a comprehensive analysis of the power dynamics involved in running ASR models on modern mobile and wearable devices.

## 3 POWER AND ACCURACY ANALYSIS OF ON-DEVICE STREAMING ASR

In this section, we apply a state-of-the-art weight pruning technique for speech recognition models, specifically Adam-pruning (Yang et al., 2022), to adjust the sizes of the Encoder, Predictor, and Joiner in on-device streaming ASR models. The details of Adam-pruning are provided in Appendix C. This approach allows us to generate a range of ASR models with varying sizes. We then analyze both the power consumption and model accuracy across these models, leading to insightful findings.

### 3.1 POWER ANALYSIS

Table 1 illustrates the characteristics of a typical on-device streaming ASR model trained on the LibriSpeech dataset (Panayotov et al., 2015), including its size, frequency of component invoca-

---

[4]Measuring the power consumption of models operating on mobile or wearable devices presents significant challenges and potential inaccuracies. Firstly, these devices may lack precise mechanisms for reporting battery levels, complicating the assessment of power usage. Secondly, while the reported battery level indicates the amount of charge remaining, it fails to provide information on voltage levels, which are crucial for accurate power measurements. Lastly, distinguishing the power consumed by the ASR model from that used by other applications or background operating system processes is particularly difficult. This complexity arises because the device's total power consumption is a cumulative effect of all active components and processes, making it hard to isolate the energy expenditure attributable solely to the model in question.

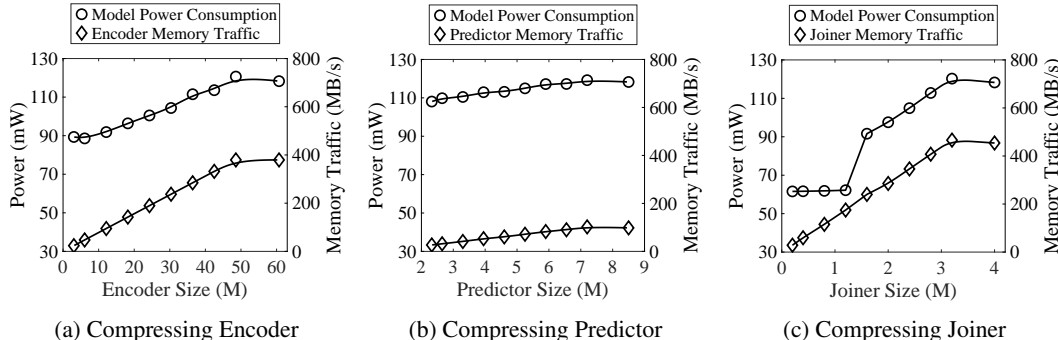

(a) Compressing Encoder      (b) Compressing Predictor      (c) Compressing Joiner

Figure 3: Models trained on LibriSpeech: Model power consumption with compressing an individual component (Encoder, Predictor, or Joiner) while keeping the sizes of the other two components constant.

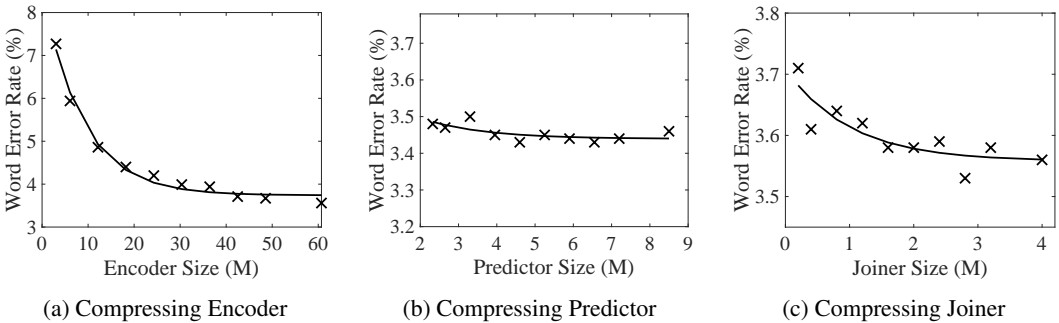

(a) Compressing Encoder      (b) Compressing Predictor      (c) Compressing Joiner

Figure 4: Models trained on LibriSpeech: Word error rate on Test-Clean with compressing an individual component (Encoder, Predictor, or Joiner) while keeping the sizes of the other two components constant. The relationship between word error rate and component size is fitted with an exponential curve.

tion, computing power, and memory power. The data reveals that computing power constitutes a surprisingly small fraction of the total power consumption (less than 1%), with memory power dominating. This high memory power primarily stems from loading model weights from memory. When comparing the model's Encoder, Predictor, and Joiner components, it is notable that even though the Encoder holds over 83% of the weights, the Joiner, with its invocation frequency being 18 times higher, generates 1.2 times more memory traffic than the Encoder. Consequently, the Joiner consumes more power. This observation contradicts the prevailing belief that larger model components use more energy, underscoring the significance of considering operational dynamics in energy optimization.

Figure 3 further explores model power consumption by compressing individual components (Encoder, Predictor, or Joiner) while maintaining the sizes of the other two components. This analysis shows that power consumption is closely tied to memory traffic, which in turn is influenced by the size of the component and how frequently it is invoked. Generally, less memory traffic results in lower power consumption. However, there is an interesting anomaly: when the Joiner is compressed to below 1.2M parameters, further reductions in its size have no effect on the model's power consumption. This plateau occurs because, at this size, the Joiner's weights fit into the energy-efficient local scratchpad memory, where data loading consumes minimal energy. This finding suggests the strategic importance of placing the weight parameters of the most energy-intensive components in local memory whenever possible, to optimize energy efficiency.

We also investigate the effects of input stride and chunk size—two key hyperparameters of streaming ASR—on the model's power consumption, revealing some interesting observations. Detailed results are provided in Appendix D.

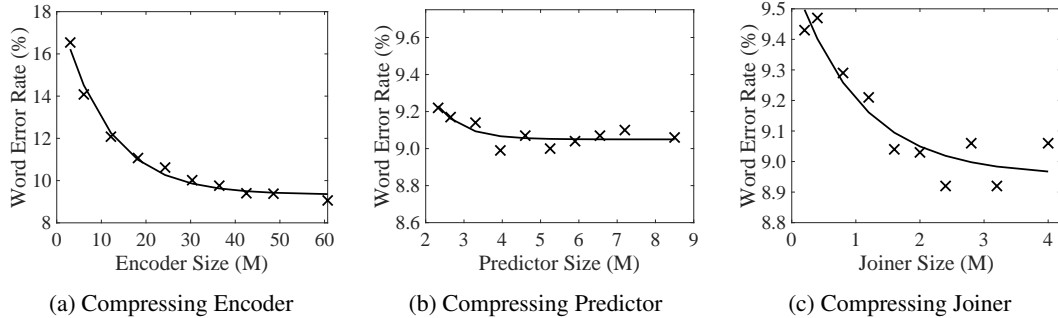

(a) Compressing Encoder      (b) Compressing Predictor      (c) Compressing Joiner

Figure 5: Models trained on LibriSpeech: Word error rate on Test-Other with compressing an individual component (Encoder, Predictor, or Joiner) while keeping the sizes of the other two components constant. The relationship between word error rate and component size is fitted with an exponential curve.

## 3.2 ACCURACY ANALYSIS

Figures 4 and 5 present the word error rates for compressed models across two evaluation sets from LibriSpeech: test-clean and test-other. It is evident that reducing the sizes of the model components typically results in an increased word error rate.[5] Among the components—Encoder, Predictor, and Joiner—the Predictor shows the least sensitivity to compression. This suggests that for energy optimization, employing a smaller Predictor, or potentially omitting it altogether, does not significantly compromise accuracy. In contrast, both the Encoder and Joiner components exhibit a notable sensitivity to compression. Interestingly, the relationship between word error rate and encoder size appears to adhere to an exponential law:

$$\text{Word Error Rate} = \exp\left(a \cdot \text{encoder\_size} + b\right) + c \tag{1}$$

By fitting the function, we determined the parameters a, b, and c, and observed a close fit between the model predictions and actual data points.

The quality of this fit is quantitatively assessed using the adjusted R-squared value (James et al., 2013), which evaluates the fit's goodness while adjusting for the number of parameters in the function to prevent overfitting from yielding artificially high values. The adjusted R-squared values obtained were 0.9832 for the test-clean set and 0.9854 for the test-other set, demonstrating a robust explanatory power of the exponential function for the observed data. This exponential relationship exists across different datasets; it also extends to models trained on Public Video, another dataset we evaluated. For further information, please see Appendix B.

This exponential relationship suggests diminishing returns with increasing encoder size, prompting the community to reconsider encoder design in on-device streaming ASR systems for a more effective balance between model size and accuracy.

We also vary the input stride and chunk size to assess their impact on model accuracy. Our observations are detailed in Appendix D.

## 4 ASR ENERGY EFFICIENCY OPTIMIZATION

Our optimization objective is to minimize the power consumption of streaming ASR models with minimal impact on their performance. We achieve this by evaluating the power and accuracy sensitivity of the Encoder, Predictor, and Joiner components. These sensitivities indicate the change

---

[5]The variability observed in the curves related to compressing the Predictor and Joiner arises from the randomness in initialization and pruning throughout training.

in power consumption and performance, respectively, for a unit reduction in component size:

$$\text{Power Sensitivity}_{\text{component}} := \frac{\Delta\text{Power}}{\Delta\text{Size}_{\text{component}}}$$

(2)

$$\text{Accuracy Sensitivity}_{\text{component}} := \frac{\Delta\text{Accuracy}}{\Delta\text{Size}_{\text{component}}}$$

Here, component refers to the Encoder, Predictor, or Joiner, and accuracy is inversely related to the word error rate.

The power consumption of on-device streaming ASR models primarily arises from loading model weights from memory. Thus, power sensitivity can be expressed as:

$$\text{Power Sensitivity}_{\text{component}} = \frac{\Delta(\text{size} \times \text{invocation frequency} \times \text{memory energy unit})}{\Delta\text{size}}$$

$$= \text{invocation frequency} \times \text{memory power unit}$$

(3)

with the memory energy unit representing the energy required to load a byte from memory. We adopt 1.5pJ/byte for SRAM and 120pJ/byte (Li et al., 2019) for off-chip memory in our study. Although not explicitly stated, power sensitivity is influenced by component size, as memory power consumption depends on whether the component's weights fit within the energy-efficient SRAM or need to be stored in the more power-intensive off-chip memory.

Accuracy sensitivity is determined by progressively reducing a component's size, observing the impact on model accuracy, and fitting a function to describe this relationship (see Equation 1). The derivative of this function quantifies accuracy sensitivity. An exponential function is used for this purpose, applied similarly across the Encoder, Predictor, and Joiner components.

Once we have both power sensitivity and accuracy sensitivity, we use their ratio—power-to-accuracy sensitivity—to guide our compression decisions:

$$\text{power-to-accuracy sensitivity ratio} = \frac{\text{power sensitivity}}{\text{accuracy sensitivity}}$$

(4)

A high power-to-accuracy sensitivity ratio indicates that compressing the component will yield the greatest power savings for a given accuracy loss. This ratio helps determine the order in which we compress components of on-device streaming ASR models.

Our compression algorithm starts with a fully uncompressed model and iteratively reduces its size to meet a user-defined power reduction target (e.g., "reduce power by 60 mW"). For each milliwatt of power reduction, we calculate the power-to-accuracy sensitivity ratio of each component and compress the one with the highest ratio. In Neural Transducer models, we observe that the Joiner initially has the highest power-to-accuracy sensitivity ratio due to its high power sensitivity, which results from its frequent invocation. Once the Joiner's size is reduced enough to fit into energy-efficient local memory, its power-to-accuracy sensitivity ratio decreases, and the predictor becomes the component with the highest ratio. The Predictor is then compressed until it reaches its user-defined minimum size, beyond which further compression would cause a significant accuracy loss due to the exponential relationship between model accuracy and component size. The Encoder is compressed next until it reaches its user-defined minimum size. If additional power reduction is needed, we return to compressing the Joiner.

Thus, the compression order follows: Joiner → Predictor → Encoder → Joiner. It's important to note that our algorithm only determines the compression order between components. Once a component is selected for compression, an existing compression algorithm handles the specific task of deciding which weight parameters to prune within that component. Our approach is compatible with any existing compression algorithm.

## 5 EXPERIMENTS

### 5.1 DATASETS AND MODELS

We experiment with two datasets: LibriSpeech and Public Video. Their details are provided in Appendix A. LibriSpeech, sourced from audiobooks, comprises 960 hours of training data and

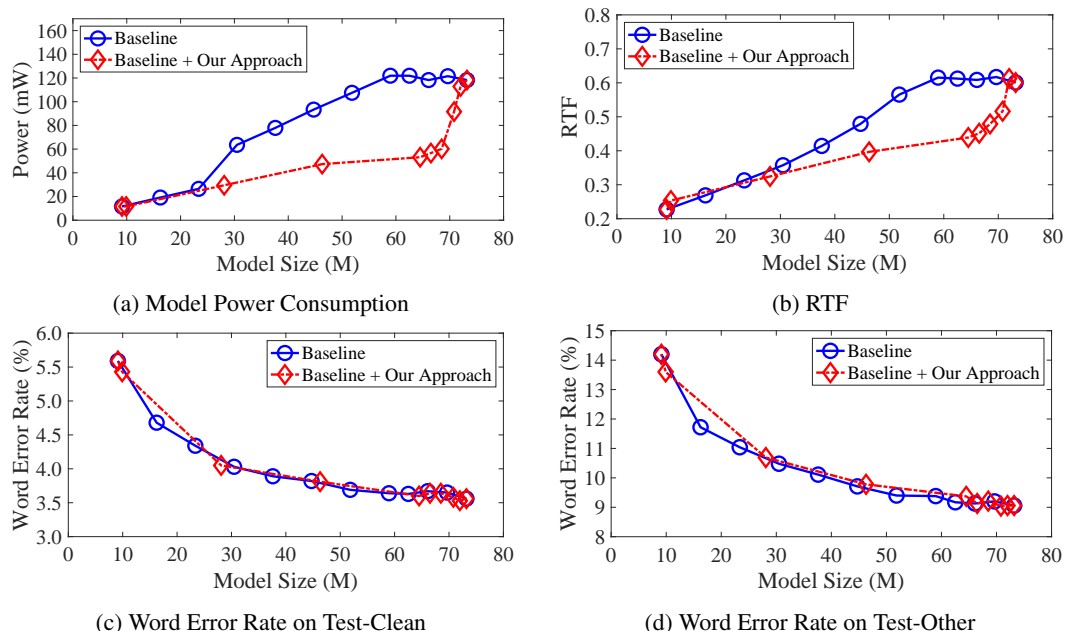

Figure 6: Models trained on LibriSpeech under different sizes and compression schemes.

includes two evaluation sets: Test-Clean, featuring easily transcribed audio recordings, and Test-Other, containing recordings challenging to transcribe due to strong speaker accents or suboptimal recording conditions. Public Video, an in-house dataset, consists of audio extracted from publicly available English videos, with the owners' consent, ensuring that data is de-identified. This dataset offers 148.9K hours of training data, along with two evaluation sets: Dictation, with 5.8K hours of unscripted, open-domain conversations, and Messaging, with 13.4K hours of audio messages.

We use Emformer models (Shi et al., 2021) with an input stride of 40ms and a chunk size of 160ms for experiments on LibriSpeech, and Conformer models (Gulati et al., 2020) with an input stride of 60ms and a chunk size of 300ms for experiments on the Public Video dataset.

## 5.2 BASELINES AND EVALUATION METHODOLOGIES

Our proposed method aims to determine which component of a multi-component model (e.g., Neural Transducer) should be prioritized for compression to achieve the most significant energy savings. Once the critical component is identified, the specific compression technique used is beyond the scope of this study; for our experiments, we utilize Adam-prune (Yang et al., 2022), a state-of-the-art compression technique for on-device streaming speech recognition.

Our comparative analysis contrasts two scenarios: one in which the baseline compression technique is uniformly applied across the entire model (referred to as "baseline"), and another in which the same baseline technique is enhanced by our approach to strategically prioritize compression (referred to as "baseline + our approach"). By comparing these scenarios, we aim to demonstrate the added value of our method, specifically highlighting the advantages of strategic component prioritization in model compression.

Although we use the strongest available baseline in our study, the choice and performance of the baseline are not critical in this context. Our focus is on demonstrating the **additional benefits** provided by our approach when combined with any baseline method. The strength of the baseline only affects the absolute accuracy of the compressed models at a target power consumption level but does not impact the additional accuracy gains achieved through our prioritization method.

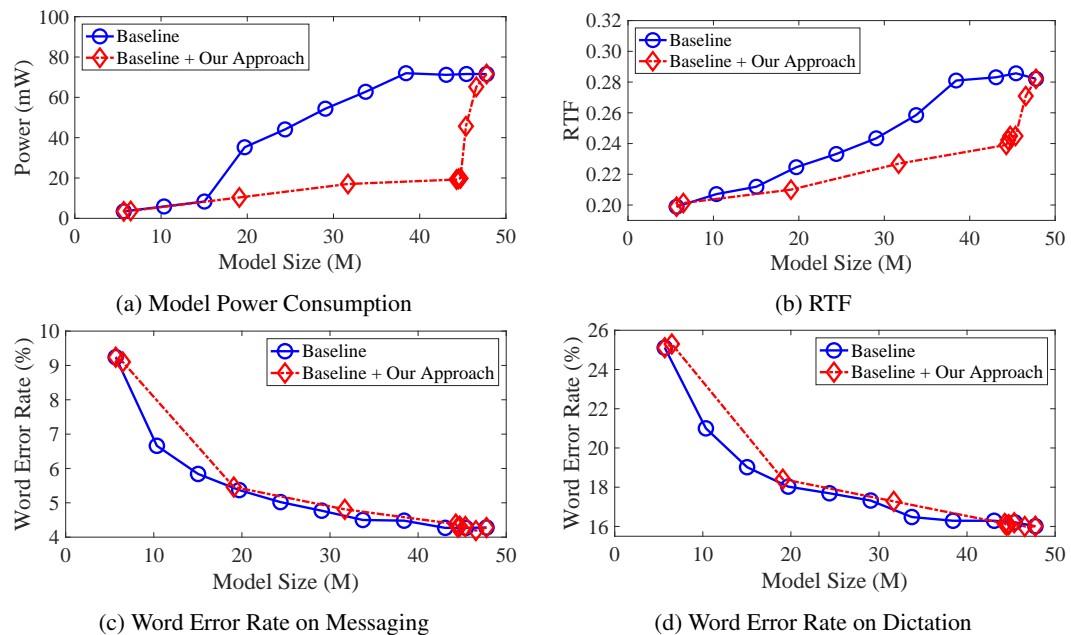

(a) Model Power Consumption

(b) RTF

(c) Word Error Rate on Messaging

(d) Word Error Rate on Dictation

Figure 7: Models trained on Public Video under different sizes and compression schemes.

## 5.3 RESULTS ON LIBRISPEECH

Figure 6 (a) illustrates the power consumption[6] across different model sizes. Our method yields a notable decrease in power usage when compared to the baseline for models ranging from 30-76 MB in size. Beyond this, further compression results in minimal size components, rendering the distinction between them less impactful. Consequently, for models under 30MB, both compression approaches exhibit comparable power consumption levels.

Figure 6 (b) presents the Real-Time Factor (RTF). Intriguingly, despite our method's focus on enhancing energy efficiency, it inadvertently improves the RTF, indicating a speedier model inference. This improvement stems from our approach's emphasis on compressing heavily utilized model components, which significantly contribute to the overall inference time. By prioritizing the compression of these components, we effectively reduce the inference time, thereby outperforming the baseline.

Figures 6 (c) and (d) detail the word error rate across various model sizes, demonstrating that our strategy preserves the baseline's model accuracy. Synthesizing the findings depicted in Figures 6 (a)–(d), our approach leads to up to a 47% reduction in energy consumption and a 29% decrease in RTF, all while maintaining comparable accuracy to the baseline.

## 5.4 RESULTS ON PUBLIC VIDEO

Figures 7 (a)–(d) present the power consumption, RTF, and accuracy for models of various sizes trained on the Public Video dataset. Our approach reduces energy consumption by up to 38% and RTF by up to 15%, all while maintaining model accuracy comparable to the baseline.

## 6 RELATED WORK

To the best of our knowledge, this study represents the pioneering effort to analyze both the operational dynamics and memory placement strategies of model components for enhancing the energy efficiency

---

[6]The power consumption is defined as the average power used by the model over the duration of incoming audio segments. In the context of streaming speech recognition, this duration is constant and unaffected by compression methods. Therefore, since energy consumption is calculated by multiplying this fixed duration by the average power consumption, it linearly correlates with the power consumption.

of on-device streaming ASR models. The most closely related works are a set of proposals on ASR model compression and energy optimization.

## 6.1 ON-DEVICE ASR MODEL COMPRESSION

Research by (Ghodsi et al., 2020) showed that removing recurrent layers from the Predictor in Neural Transducer models does not negatively impact word-error rates. This discovery highlights the potential for both compressing these models and enabling the Predictor to operate statelessly, given its crucial function in reducing repetitive outputs. Further explorations by (Botros et al., 2021) aimed at reducing the footprint of the Predictor and Joiner components in Neural Transducer models to boost processing efficiency. They explored parameter sharing between the embedding matrices of the Predictor and Joiner, suggesting a weighted-average embedding to encapsulate the history of the Predictor's tokens. (Shangguan et al., 2019) proposed shrinking the Predictor by replacing its Long Short-Term Memory (LSTM) units with Simple Recurrent Units (SRU) that include 30% structured sparsity. They also recommended adapting the Encoder with Coupled Input-Forget Gate (CIFG) LSTM variants that include 50% structured sparsity. (Yang et al., 2022) employed a Supernet-based neural architecture search to determine optimal sparsity levels for each layer, aiming to balance model accuracy and size. While these studies aimed at reducing model size or RTF without significantly compromising accuracy, they did not explore how to reduce power consumption, which is our primary interest.

## 6.2 ON-DEVICE ASR ENERGY OPTIMIZATION

Prior initiatives to lower the Neural Transducer's power consumption have concentrated on modifying the model's cell architecture. (Li et al., 2024) introduced folding attention, achieving a reduction in both model size and power consumption by 24% and 23%, respectively, without detriment to accuracy. (Venkatesh et al., 2021) streamlined LSTM cells and designed a deeper yet narrower Neural Transducer model, cutting down off-chip memory access by 4.5 times and energy costs by twice, with only a minor accuracy decrease. Our research differs from these methods by focusing on understanding the runtime behaviors of Neural Transducer model components. This understanding aids in directing compression strategies more effectively toward energy optimization.

## 7 CONCLUSION

Power consumption remains a critical challenge for on-device streaming ASR, directly affecting device recharge frequency and user experience. This study conducted extensive experiments to analyze power usage in ASR models, examining its correlation with model runtime behaviors and identifying strategies for power reduction. Our findings highlight that the majority of ASR power consumption is attributed to loading model weights from off-chip memory, intricately linked to the size of model components, their invocation frequency, and their memory placement. Interestingly, despite its smaller size, the Joiner component consumes more power than the Encoder and Predictor, due to these factors. Additionally, we discovered a notable exponential relationship between the model's word error rate and the encoder size. Utilizing these insights, we formulated a series of design guidelines focused on model compression for enhancing energy efficiency. The application of these guidelines on the LibriSpeech and Public Video datasets resulted in significant energy savings of up to 47% and a reduction in RTF by up to 29%, all while preserving model accuracy compared to the state-of-the-art methods. These outcomes underscore the potential of targeted model optimization strategies in achieving substantial energy efficiency improvements, marking a pivotal step towards sustainable and user-friendly on-device streaming ASR technologies.

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

## A  DETAILS OF THE DATASETS

### A.1  LIBRISPEECH

LibriSpeech Panayotov et al. (2015), is a prominent corpus extensively utilized in speech recognition research. This corpus features 960 hours of English speech, sourced from audiobooks available through the LibriVox project, which are in the public domain. It includes two main evaluation sets tailored for different testing scenarios:

- Test-Clean: This subset consists of high-quality, clean audio recordings. It provides an ideal condition for benchmarking the baseline performance of speech recognition systems due to its clarity and ease of transcription.
- Test-Others: This subset encompasses recordings that present a variety of challenges, such as accents, background noises, and lower recording qualities. It serves as a stringent testing environment to evaluate the robustness and adaptability of speech recognition technologies under less-than-ideal conditions.

### A.2  PUBLIC VIDEO

The Public Video dataset, an in-house collection, is derived from 29.8K hours of audio extracted from English public videos. This dataset has been ethically curated with the consent of video owners and further processed to ensure privacy and enhance quality. We de-identify the audio, aggregate it, remove personally identifiable information (PII), and add simulated reverberation. We further augment the audio with sampled additive background noise extracted from publicly available videos. Speed perturbations Ko et al. (2015) are applied to create two additional copies of the training dataset at 0.9 and 1.1 times the original speed. We apply distortion and additive noise to the speed-perturbed data. These processing steps eventually result in a total of 148.9K hours of training data. For evaluating the performance of models trained on this dataset, we use the following two test sets:

- Dictation: This subset consists of 5.8K hours of human-transcribed, anonymized utterances, sourced from a vendor. Participants were asked to engage in unscripted open-domain dictation conversations, recorded across various signal-to-noise ratios (SNR), providing a diverse assessment environment.

- Messaging: This subset comprises 13.4K hours of utterances, sourced from a vendor. It features audio messages recorded by individuals following scripted scenarios intended for an unspecified recipient. These utterances are generally shorter and incorporate more noise than those in the dictation subset, offering a different dimension to evaluate ASR systems.

## B    ACCURACY OF ASR MODELS TRAINED ON PUBLIC VIDEO

We applied compression to the Encoder of the ASR model trained using the Public Video dataset. The impact of this compression on word error rates across two evaluation sets, Dictation and Messaging, is depicted in Figures 8 (a) and (b). To analyze the data, we employed the function outlined in Equation 1, which proved to be an excellent fit; the predictions derived from this function align closely with the observed data. Quantitatively speaking, the adjusted R-squared values—0.9760 for Dictation and 0.9851 for Messaging—underscore the exponential relationship between word error rate and encoder size, reaffirming this pattern's consistency across different datasets.

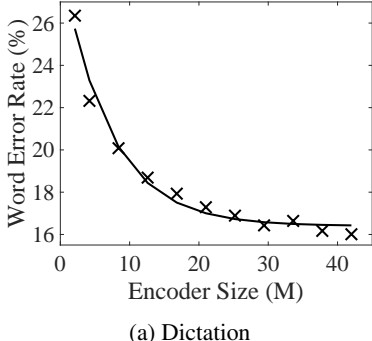 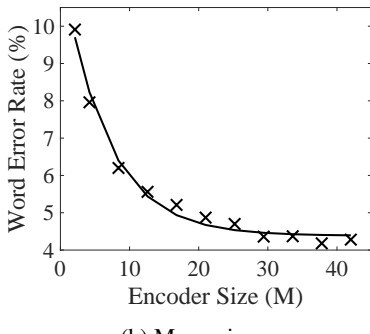

(a) Dictation                                    (b) Messaging

Figure 8: Models trained on the Public Video dataset: Word error rate with compressing Encoder while keeping the size of Predictor and Joiner. The relationship between word error rate and component size is fitted with an exponential curve.

## C    DETAILS OF THE ADAM-PRUNING COMPRESSION ALGORITHM

Adam-pruning is an iterative method designed to prune a model or its components. Each pruning step is executed over $N$ training epochs. During each step, Adam-pruning evaluates the square of the gradient ($E\left[\left(\frac{\partial l}{\partial w}\right)^2\right]$) for every non-sparse parameter $w$ in the model. A larger square of the gradient suggests that pruning the parameter would result in a substantial change in the model's performance. Based on this, Adam-pruning prunes only the parameters with the top $K$ smallest gradient squares at the end of each pruning step. After $M$ such steps, Adam-pruning reduces the model to a desired level of sparsity.

## D    IMPACT OF INPUT STRIDE AND CHUNK SIZE ON MODEL ACCURACY AND POWER CONSUMPTION

Input stride and chunk size are two essential hyperparameters for on-device streaming ASR. Input stride defines the time window over which input frames are combined into an aggregated frame that is then fed into the model. Chunk size refers to the time duration over which these aggregated frames are processed together as a batch by the model. In this section, we explore the effects of varying input stride and chunk size on both dense and sparse models.

We first vary the input stride from 20 milliseconds to 40 milliseconds and evaluate the accuracy and power consumption of four models trained on LibriSpeech: a dense model, a model with 80% sparsity in its encoder, a model with 80% sparsity in its predictor, and a model with 80% sparsity in its joiner. The results are provided in Tables 2 and 3. Our findings are as follows:

Table 2: Impact of input stride on the model accuracy trained on LibriSpeech.

| Word Error Rate (%) | Input Stride | Dense Model | 80% Sparse Encoder | 80% Sparse Predictor | 80% Sparse Joiner |
|---|---|---|---|---|---|
| Test-Clean | 20ms | 3.61 | 4.72 | 3.61 | 4.17 |
| | 40ms | 3.56 | 4.86 | 3.60 | 3.64 |
| Test-Other | 20ms | 9.13 | 11.90 | 9.13 | 9.58 |
| | 40ms | 9.06 | 12.08 | 9.14 | 9.29 |

Table 3: Impact of input stride on the power consumption of models trained on LibriSpeech.

| Model Power Consumption (mW) | Input Stride | Dense Model | 80% Sparse Encoder | 80% Sparse Predictor | 80% Sparse Joiner |
|---|---|---|---|---|---|
| | 20ms | 131 | 104 | 123 | 62 |
| | 40ms | 118 | 92 | 110 | 62 |

- Observation 1: A smaller stride can have both positive and negative effects on model performance.

- Observation 2: A smaller stride generally increases power consumption.

Regarding the first observation, input stride is used to enhance training and inference efficiency by reducing sequence length. While a smaller stride better preserves acoustic local features, which typically improves performance, it can also introduce risks such as greater sensitivity to noise and loss of broader contextual information. A stride of 4–6 is commonly chosen as it balanced accuracy and efficiency.

As for the second observation, in streaming ASR, a smaller stride increases the number of segments, resulting in more frequent decoding of blank tokens and thus more frequent invocation of the joiner, which raises power consumption. However, if the joiner is compressed to fit within the local SRAM, this increased invocation has minimal impact on power usage, due to the high energy efficiency of SRAM.

We also vary the chunk size from 160ms to 320ms and measure the accuracy and power consumption of four models: a dense model, a model with 80% sparsity in its encoder, a model with 80% sparsity in its predictor, and a model with 80% sparsity in its joiner. The results are provided in Tables 4 and 5. Our observations are as follows:

- Observation 3: Increasing the chunk size generally improves model accuracy.

- Observation 4: Larger chunk sizes reduce model power consumption.

Table 4: Impact of chunk size on the model accuracy trained on LibriSpeech.

| Word Error Rate (%) | Chunk Size | Dense Model | 80% Sparse Encoder | 80% Sparse Predictor | 80% Sparse Joiner |
|---|---|---|---|---|---|
| Test-Clean | 160ms | 3.56 | 4.86 | 3.60 | 3.64 |
| | 320ms | 3.50 | 4.60 | 3.50 | 3.52 |
| Test-Other | 160ms | 9.06 | 12.08 | 9.14 | 9.29 |
| | 320ms | 8.82 | 11.75 | 8.83 | 8.90 |

Table 5: Impact of chunk size on the power consumption of models trained on LibriSpeech.

| Model Power Consumption (mW) | Chunk Size | Dense Model | 80% Sparse Encoder | 80% Sparse Predictor | 80% Sparse Joiner |
|---|---|---|---|---|---|
| | 160ms | 118 | 92 | 110 | 62 |
| | 320ms | 94 | 86 | 87 | 38 |

For the third observation, larger chunk sizes enable the encoder to capture relationships between segments more effectively, improving performance. However, smaller chunk sizes have the advantage of lowering decoding latency.

As for the fourth observation, in streaming ASR, a larger chunk size decreases the frequency at which the encoder is invoked, thereby reducing memory power usage and overall power consumption.

