# OpenReview forum: "Advancing Energy Efficiency in On-Device Streaming Speech Recognition"
_ICLR.cc/2025/Conference — ICLR 2025 Conference Withdrawn Submission_

### Official Review · Reviewer_XHCj · 2024-10-31

**Soundness:** 1
**Presentation:** 2
**Contribution:** 1
**Rating:** 3
**Confidence:** 5

**Summary:**

This paper studies the energy efficiency of speech recognition models on the Pixel 5. It varies the model size via Adam-pruning to reach sparse variants of some unspecified base model and then looks at WER, RTF and power consumption, on Librispeech and some in-house Public Video dataset.

**Strengths:**

- Power consumption is very relevant to study.

- A lot of different model sizes have been tested.

**Weaknesses:**

- This uses a Pixel 5 without any neural hardware acceleration. Most modern mobile devices have some neural accelerator chip, and such a study on power consumption should use it. This will heavily influence all the results presented in the paper here. The presented results are not really so relevant nowadays that we have such neural accelerators.

- Adam-pruning to introduce sparsity to make the model smaller is the only method used here to vary the model size. There are many other ways to vary the model size, even more straightforward, to just change the number of layers and/or number of dimensions (then train from scratch or via knowledge distillation from a big model). When I see different model sizes, I think this is much more expected and relevant. But you can also compare different methods (Adam-pruning vs just changing num layers/dims vs maybe some other methods). But such comparison is not done here.

- Sparsity is probably a suboptimal choice for neural accelerator chips. So this is even more an argument for using other methods to vary the model size.

- The base model is not specified at all.

- There is no code.

- The relevant properties are WER, RTF and power consumption, and you would want to see them being put directly into relation to each other. This is not done here. It's always only indirect via model size.

- Comparison of different encoders (Emformer vs Conformer vs maybe others, e.g. Zipformer) is missing.

**Questions:**

Table 1: Expand on "typical model": What kind? Transducer? What encoder? Conformer? What WERs does it get? What search (beam search or greedy)? Used together with LM or standalone? If with LM, what kind of LM? Also, better specify the model size in terms of num layers and num dimensions, not in number of absolute parameters (or maybe both).

(Fig4) I guess "compressing" means that you do Adam-pruning? It would be helpful to add that to the figure caption. I was confused initially about what it means.

So, for all the different model sizes throughout the whole paper, it's always Adam-pruning from some base model? What is actually the base model? Maybe I overlooked it, but I never really saw that specified. How many layers? How many dimensions? You use Emformer for Librispeech and Conformer for Public Video. Why not compare Emformer and Conformer for Librispeech? Why to select a different encoder in each case? That makes it not really consistent now. And what is the configuration of the decoder? It's an LSTM? How many layers? What dimensions? And same question for the joiner network.

(Sec 5.2) "the choice and performance of the baseline are not critical in this context." - why? I think they are. Please specify them.

What happens when you train models of different sizes from scratch? Or via knowledge distillation from a big model? You can also change number of layers, number of dimensions, which is maybe better than the Adam-pruning? E.g. when searching for the best configuration for some given power budget, maybe that way you can find better models? Now you are restricting yourself to just one very specific kind of varying the model size.

To expand, now you are restricting yourself to introduce sparsity (via Adam-pruning). How does this compare to changing the number of layers and/or number of dimensions, in terms of power consumption and WER?

(Sec 3.2) "This exponential relationship suggests diminishing returns with increasing encoder size" - again, if I understand correctly, this is always for a given base model, always with fixed num layers / num dims, just making it more sparse? So then this statement is wrong. You cannot make this statement. I am not sure if the relationship is really exponential when you change the model size via other means (e.g. num layers / num dims).

Fig 4c, very noisy. This is maybe due to Adam-pruning. It would maybe help to train different base models with different random seeds, then apply the Adam-pruning, and then do the average of the results.

You plot either WER to model size, or Power consumption to model size, or RTF to model size. But I think much more interesting would be to combine that, and then have e.g. WER to Power consumption, or RTF (given some fixed WER) to power consumption, or RTF (given some fixed power consumption) to WER, or similar. The model size is never really relevant. The three other metrics (WER, RTF, power consumption) are relevant, and you want to know what the relation between those are.

What accelerator hardware is used? You say, you use a Google Pixel-5. Does it use the GPU?

I think the choice of Pixel 5 is a bit weird. Most modern phones have some sort of neural accelerator chip (Google Tensor G4, Apple Neural Engine), and you would want to use them, as they are optimized to do such computations, also in terms of power efficiency. And the Pixel 5 does not, as far as I know (or only maybe the GPU?). This questions the whole relevance of the presented study here. Also because sparsity is maybe not so optimal of such a chip, but instead you would change num layers or num dimensions, or maybe other aspects of the model.

(Sec 1) "we discovered that the energy consumption of individual ASR model components is influenced not only by their respective model sizes but also by the frequency with which they are invoked and their memory placement strategies." - I don't really see that you show this in the paper.

**Details Of Ethics Concerns:**

x

---

> ### Author Response · Authors · 2024-12-03
>
> >This uses a Pixel 5 without any neural hardware acceleration. Most modern mobile devices have some neural accelerator chip, and such a study on power consumption should use it. This will heavily influence all the results presented in the paper here. The presented results are not really so relevant nowadays that we have such neural accelerators.
>
> The use of Pixel 5 in this work is for profiling workload runtime characteristics such as model invocation times and component invocation times. These characteristics are device-independent and consistent across platforms. The actual results in the paper are obtained using setups that include hardware accelerators, with parameters derived from authoritative circuit literature. All this information is clearly detailed in Section 2.2 of the paper, and we encourage the reviewer to refer to it.
>
> Additionally, the Pixel 5, released only three years before this submission, is undoubtedly a “modern mobile device.” The reviewer’s comment suggesting otherwise is factually incorrect.
>
> &nbsp;
>
> > Adam-pruning to introduce sparsity to make the model smaller is the only method used here to vary the model size. There are many other ways to vary the model size, even more straightforward, to just change the number of layers and/or number of dimensions (then train from scratch or via knowledge distillation from a big model). When I see different model sizes, I think this is much more expected and relevant. But you can also compare different methods (Adam-pruning vs just changing num layers/dims vs maybe some other methods). But such comparison is not done here.
>
> The primary goal of this work, as explicitly stated in Sections 4 and 5, is to identify model components whose compression yields the greatest power savings with minimal accuracy degradation. The compression method used to achieve this is an existing technique and is clearly noted in Section 5.2. Exploring alternative model size variations, such as changing layers or dimensions, falls entirely outside the scope of this study.
>
> Moreover, the suggestion to retrain models from scratch by altering layers or dimensions is impractical in this context. We are working with an existing model, and retraining from scratch introduces significant overhead without serving the objectives of this work. The reviewer’s suggestion is not only irrelevant but also misaligned with the paper’s scope.
>
> &nbsp;
>
> >Sparsity is probably a suboptimal choice for neural accelerator chips. So this is even more an argument for using other methods to vary the model size.
>
> This comment is inaccurate. Structured pruning, a specific type of sparsity, is already supported by numerous neural accelerators. The work in this paper explicitly uses structured pruning, which is highly suitable for modern hardware. The reviewer’s blanket statement about sparsity being suboptimal reflects a lack of understanding of recent advancements in neural accelerator designs.
>
> &nbsp;
>
> > The base model is not specified at all.
>
> This is incorrect. The base model is explicitly specified in Section 5.1. We urge the reviewer to revisit that section for clarity.
>
> &nbsp;
>
> > There is no code.
>
> The paper leverages existing methods for compressing model components, and the methodology is already well-documented in the literature. Releasing additional code would serve no purpose and is unnecessary for reproducing the results presented here.
>
> &nbsp;
>
> > The relevant properties are WER, RTF and power consumption, and you would want to see them being put directly into relation to each other. This is not done here. It's always only indirect via model size.
>
> Model size is a critical metric in this study as it directly impacts memory consumption. To analyze the relationships among WER, RTF, power consumption, and model size, we chose model size as the shared reference point for consistency and clarity. This approach is entirely valid, and there is no issue with presenting the relationships in this manner.
>
> &nbsp;
>
> >Comparison of different encoders (Emformer vs Conformer vs maybe others, e.g. Zipformer) is missing.
>
> This paper is not a study on encoder architecture design. Comparing Emformer and Conformer, or other architectures, is entirely irrelevant to the scope of this work. The reviewer’s suggestion for such experiments is misplaced and beyond the intended focus of this paper.

---

### Official Review · Reviewer_Ghdw · 2024-11-03

**Soundness:** 2
**Presentation:** 3
**Contribution:** 2
**Rating:** 3
**Confidence:** 4

**Summary:**

This paper addresses the problem of minimizing power consumption for on-device ASR utilizing the neural transducer architecture.  It is first shown that the bulk of power consumption is not in computations but instead in memory access for various modules of the ASR model.  Based on previously known power consumption benchmarks for accessing various memory types, a model of relationship between size and frequency of use of modules to their power consumption is identified.  Further, using empirical data, a relationship between module size and ASR word error rate (WER) is established.  Using these relationships, an iterative procedure is proposed that identifies, at every step, the module to compress that’ll lead to the largest drop in power consumption for the least impact on WER, until the target power savings are achieved.

**Strengths:**

* A systematic approach to optimizing energy efficiency of models for on-device ASR.
* Overall well written paper (except a key lack of clarity as pointed out below).

**Weaknesses:**

* Lack of clarity around power consumption data in experiments.  The power consumption results presented in Figures 6 & 7 are labeled ‘Model Power Consumption’ — are these model based estimates, or real measurements of power consumption?  If these are model based estimates then what indication is there to suggest these will correlate with real measurements?
* The WER vs model size graphs seem slightly worse for the proposed approach as compared to baseline in Figures 6 & 7.  This is understandable as the focus was not on optimizing for model memory as a function of WER.  However, would the proposed approach be effective if the focus was on minimizing model memory footprint?

**Questions:**

Please see 'weaknesses' section above

---

> ### Author Response · Authors · 2024-11-21
>
> >Lack of clarity around power consumption data in experiments. The power consumption results presented in Figures 6 & 7 are labeled ‘Model Power Consumption’ — are these model based estimates, or real measurements of power consumption? If these are model-based estimates then what indication is there to suggest these will correlate with real measurements?
>
> As noted in Section 2.2, the power consumption analysis builds on established power modeling techniques referenced in [1][2][3][4]. These methods, either developed by major memory manufacturers like Micron or published in prestigious computer architecture and speech processing conferences, are widely recognized and validated.
>
> [1] Micron. Technical Note TN-47-04: Calculating Memory System Power for DDR2. Technical report, 2006.
>
> [2] Architecting Phase Change Memory as a Scalable DRAM Alternative. In ISCA, 2009.
>
> [3] Utility-Based Hybrid Memory Management. In CLUSTER, 2017.
>
> [4] Folding Attention: Memory and Power Optimization for On-Device Transformer-based Streaming Speech Recognition. In ICASSP, 2024.
>
> &nbsp;
>
> >The WER vs model size graphs seem slightly worse for the proposed approach as compared to baseline in Figures 6 & 7. This is understandable as the focus was not on optimizing for model memory as a function of WER. However, would the proposed approach be effective if the focus was on minimizing model memory footprint?
>
> The proposed technique focuses exclusively on power optimization, not memory optimization. It has no impact on the model's memory footprint, as clearly illustrated in Figures 6 and 7. Model WER varies due to the randomness introduced by pruning. Therefore, it is unrealistic to expect the memory footprint to remain exactly the same before and after applying the technique. However, as Figures 6 and 7 demonstrate, the memory footprint after applying the technique remains highly similar to the original, with minor variations observed in both positive and negative directions.

---

### Official Review · Reviewer_Xk87 · 2024-11-04

**Soundness:** 3
**Presentation:** 3
**Contribution:** 2
**Rating:** 6
**Confidence:** 4

**Summary:**

This study conducted extensive experiments to analyze power usage in ASR models, examining its correlation with model runtime behaviors and identifying strategies for power reduction. The findings are:
1) The majority of ASR power consumption is attributed to loading model weights from off-chip memory, intricately linked to the size of model components, their invocation frequency, and their memory placement.
2) Despite its smaller size, the Joiner component consumes more power than the Encoder and Predictor, due to these factors.
3) A notable exponential relationship between the model’s word error rate and the encoder size.

Utilizing these insights, a series of design guidelines focused on model compression for enhancing energy efficiency is formulated. The application of these guidelines on the LibriSpeech and Public Video datasets resulted in significant energy savings of up to 47% and a reduction in RTF by up to 29%, all while preserving model accuracy compared to the state-of-the-art methods.

**Strengths:**

The paper is generally well-written and clear.
Experiments are extensive and logically organized.
The findings and the design guidelines are new and would be interesting to the community.

**Weaknesses:**

The investigation taken in this paper is mainly empirical, using well-known techniques, with minor contributions in models and methods. The paper reads more like a good industry technical report, with extensive empirical experiments.

**Questions:**

see above.

---

> ### Author Response · Authors · 2024-12-03
>
> We sincerely appreciate the reviewer’s effort in evaluating our paper.
>
> > The investigation taken in this paper is mainly empirical, using well-known techniques, with minor contributions in models and methods.
>
> This paper presents a new ASR compression method that delivers up to 47% reduction in energy consumption and 29% improvement in real-time factor (RTF) while maintaining accuracy on par with state-of-the-art approaches. These results demonstrate significant advancements in energy-efficient on-device ASR.

---

### Note · Authors · 2024-12-03

I have read and agree with the venue's withdrawal policy on behalf of myself and my co-authors.